# Examining the Impact of Stressors during COVID-19 on Emergency Department Healthcare Workers: An International Perspective

**DOI:** 10.3390/ijerph19063730

**Published:** 2022-03-21

**Authors:** Elizabeth Keller, Meghan Widestrom, Jory Gould, Runcheng Fang, Kermit G. Davis, Gordon Lee Gillespie

**Affiliations:** 1College of Nursing, University of Cincinnati, Cincinnati, OH 45221, USA; kellere4@mail.uc.edu; 2Department of Environmental and Public Health Sciences, University of Cincinnati, Cincinnati, OH 45221, USA; wellermn@mail.uc.edu (M.W.); gouldjo@mail.uc.edu (J.G.); fangrg@mail.uc.edu (R.F.); daviskg@ucmail.uc.edu (K.G.D.)

**Keywords:** healthcare, stress, burnout, wellbeing, global workforce

## Abstract

Emergency department healthcare workers are known to face a unique combination of pressures from their careers and work environments regularly. Caring for dying patients and making difficult lifesaving decisions not only continued but also became more prevalent for emergency department healthcare workers during the COVID-19 pandemic. A growing body of literature revealed that the mental and emotional toll of COVID-19 has been tremendous. However, the burden of COVID-19 on the overall physical health and work–life balance on this group needs to be understood. This study aimed to describe the impact of stress on wellbeing and health across the globe among emergency department healthcare workers. A cross-sectional survey comprising work–family and family–work conflict scale, work–life balance, physical symptoms inventory, Oldenburg Burnout Inventory, satisfaction with job and life, and life change index scale was distributed to a convenience sample through listservs and social media. In total, 287 participants responded, 109 completing all questions. Fatigue was the most common symptom reported to occur daily (28.4%, *n* = 31), followed by muscle pain (13.8%, *n* = 15) and backache (11.9%, *n* = 13). Nurse practitioners reported the highest number of physical symptoms and the highest average scores and counts of stressful life events, while registered nurses indicated the highest work–family conflict levels. Linear regressions showed that stressful life events are significantly associated with both physical symptoms and work–family conflict. Results underscore the need to better support emergency department workers to mitigate the risks associated with occupational stress. Protective organizational policies and increased support strategies may be employed to improve wellbeing and cultivate a more sustainable workforce.

## 1. Introduction

Over two years have passed since the COVID-19 pandemic swept across the globe, bringing with it death, despair, and an overcrowding in hospitals and emergency departments. As of 13 March 2022, the World Health Organization (2022) reported over 455 million cases of COVID-19 worldwide, and over 6.0 million deaths globally [1]. The United States remains the country with the largest global outbreak, accumulating over 79 million confirmed cases and 971,162 deaths [2].

The proliferation of cases paired with new waves and variants of SARS-CoV-2, the virus that causes COVID-19, has placed unprecedented pressure on healthcare systems. Healthcare workers remain under stressful working conditions and must fight against fatigue as the pandemic persists. Many professionals worked continuously for more than 100 days, negatively affecting their sleep and subsequent patient outcomes [3,4]. While hospitals implemented many protections for the severe acute respiratory syndrome of COVID-19, many of these measures were astringent and added to the stress of the healthcare workers [5]. During the COVID-19 pandemic, stressors experienced by health care professionals have been reported as limited resources at their place of work, the threat of exposure to SARS-CoV-2, long shift hours, personal ethical dilemmas regarding fear of exposing their family to the virus, neglect for personal needs, and lack of information [6,7]. Mandatory policies of wearing respirators and other personal protective equipment may have increased the stress, especially give the uncomfortable nature of wearing them long-term and continuously [8]. This growing body of scientific literature has revealed that the mental and emotional toll of COVID-19 has been tremendous. Workers on the front-line have faced, and continue to face, enormous mental stress because of prolonged overloaded work and witnessing many deaths, leading to sadness and frustration [9,10].

Even before the pandemic, research has shown that emergency department healthcare workers tend to face a unique combination of pressures from their careers and work environments, where the balance of life and death rests in their decisions [11]. Characteristics of burnout through reports of emotional exhaustion and depersonalization are often high among emergency healthcare workers, but their levels tend to significantly differ between roles [12]. For example, Schooley and associates [12] revealed that emergency department nurses had lower overall emotional exhaustion when compared to medical technicians, yet physicians had overall lower depersonalization when compared to nurses and medical technicians.

As the pandemic progressed, a study was conducted in Wuhan, China that investigated the work stress levels of nurses working and responding to the surges of cases of COVID-19 in emergency departments [13]. Using the Chinese version of the Stress Overload Scale and the Self-Rating Anxiety Scale, they found a positive correlation between stress and anxiety. Results suggested that nurses who worked at the stretcher side during the COVID-19 pandemic were under immense pressure. Similarly, a Michigan-based study used a qualitative questionnaire with nurses and determined that 50% of the participants experienced symptoms of depression and anxiety, one-third of participants had symptoms of posttraumatic stress disorder (PTSD), and more than 85% expressed fear of going to their workplace [14]. Raudenská and colleagues (2020) reported similar findings among physicians, revealing how stress increased their risk for emotional trauma, acute stress disorder, PTSD, and burnout related to working during the pandemic [7]. Basically, the COVID-19 pandemic has placed a tremendous strain on healthcare workers, given the intense work demands that resulted from high volumes of patients with a highly contagious and deadly virus that results in high mortality [15].

Despite the current body of evidence, few studies have examined and distinguished between the various emergency department roles together, or roles beyond registered nurses or physicians. Therefore, the overall physical health and work–life balance on those working in the emergency department need to be better understood. Understanding, defining, and measuring wellbeing in the workplace is a growing field of research stemming from Total Worker Health^®^ (Atlanta, GA, USA) initiatives [16,17]. Thus, it is essential to describe the impact of stressful events and any spill-over effects on worker wellbeing, captured by physical health and emotional impact through work–family conflict.

The purpose of this study was to describe the stressful events that occurred during the COVID-19 pandemic, and how wellbeing was impacted for emergency department healthcare workers with varying roles and regions of the world in which they live. This analysis focused on the relationship of stress with physical symptoms and work–family conflict. The results may provide understanding and reference for the experiences of the different emergency department roles, and for where future interventions can be targeted to ensure adequate resources and protective organizational policies are in place. This in turn will support a sustainable workforce, which indirectly serves to improve overall patient outcomes.

## 2. Materials and Methods

### 2.1. Design and Measures

This research used a correlational, cross-sectional design. A quantitative survey was developed using Total Worker Health^®^ (Atlanta, GA, USA) surveys. The survey included a total of 97 questions and was distributed in English. The survey included screening questions regarding eligibility, work–family and family–work conflict scale [18], work–life balance and satisfaction with job and life [19], physical symptoms inventory [20], Oldenburg Burnout Inventory [21], life change index scale (the stress test) [22], and demographic questions. Survey questions can be found in the Appendix A. The University of Cincinnati Institutional Review Board (IRB) approved the study and study documents. Before each participant began the survey, they were presented with an information sheet for research that they had to review and agree to, to participate before completing the survey.

### 2.2. Sample and Recruitment

The population of interest included emergency department healthcare workers working during the COVID-19 pandemic across the globe from April to August 2021. Inclusion criteria were those aged 18 or older, working in the emergency department at the bedside delivering patient care, and with the ability to read English (at 8th grade level).

Emergency department healthcare workers were sampled through convenience and snowball sampling methods. Friends and colleagues of the research team were contacted to help distribute the surveys via email and social media and encouraged participants to share with other eligible coworkers. The opportunity to participate in this research study was also shared on social media outlets, such as Twitter, through direct messaging public profiles with “emergency department” listed in their biographical sections.

Upward of 400 participants were invited to take part in the study, without restriction to country, although we cannot be sure how many participants our invitation reached due to the snowballing approach where the invitation could be shared by colleagues. A total of 287 participants responded to the survey through a REDCap secure link at their own convenience. However, 62 participants did not meet the inclusion criteria by either being under 18 years old or because they were not currently working during the pandemic. Further, 81 participants did not agree to participate, and 35 had missing quantitative responses. In total, 109 participants completed all 5 survey elements that were required for data analysis. See Figure 1 for a flowchart of the study enrollment procedures.

### 2.3. Surveys

Stressors of the participants in this study were determined using the life change index scale (also known as the stress test). This test asks about 43 different events that may have occurred within the past year or would happen in the near future. This measure determines both the life change index score (where events are correlated with an impact score and then summed) and the life change index count (summing the frequency of events). The life change index score gives more stressful events (e.g., death of a spouse, death of family member, personal injury) more weight than less stressful events (e.g., change in sleeping or eating habits). The life change index count helps to identify the number of total events which the participant had experienced. Both measures provide insight into the population, as the life change index score highlights the intensity of events, whereas the life change index count values the number of experienced events. Because the life change index scale is used to determine the change in stimuli to participants’ bodies and subsequent stress, this scale helps to determine the likeliness of illness in the near future. The greater the life change index score and life change index count, the harder it is to return to good health, and the higher the likelihood of experiencing illness soon.

Participants were asked four questions related to their work–family conflict (or work–life balance) that they could agree or disagree with. Their results were summed to generate a score from four (indicating good work life balance and no work–family conflict) to 20 (indicating a lack of work–life balance and increase in work–family conflict).

Participants were asked to report if they experienced any of the following physical symptoms thought to be associated with psychological stress at least once or twice over the past month: upset stomach or nausea, backache, headache, acid indigestions or heartburn, diarrhea, stomach cramps, loss of appetite, shortness of breath/difficulty breathing, dizziness, chest pain, flu or cold symptoms, muscle pain, and tiredness or fatigue. According to the scale, items were rated in frequency from 1 = not at all to 5 = every day. Results were summed for a response ranging from 13 (no reported physical symptoms) to 65 (high number of reported physical symptoms), according to the instructions of the measurement tool.

### 2.4. Statistical Analysis

Descriptive statistics were computed for the variables of stressful life events, physical symptoms, and work–family conflict for analysis to explore the research purpose. Stressful life events served as the independent variable (measured both as a sum and a count). A sum of the physical symptoms and sum of work–family conflict served as dependent variables. Linear regressions were used to determine the impact of stress on physical health and work–family conflict.

## 3. Results

### 3.1. Sociodemographic and Descriptive Statistics

Data were elicited from 15 countries (see Table 1 for detailed demographics). Responses were grouped by region for analysis as North America (including Canada, Mexico, and the United States; *n* = 61, 62.8%), Asia (including China, Turkey, India, Pakistan, Philippines, Qatar, Saudi Arabi; *n* = 20, 20.6%), Europe (Slovenia and the United Kingdom; *n* = 11, 11.3%), South America (including Columbia; *n* = 1, 1.0%), Australia (*n* = 4, 4.12%), and Africa (including Tanzania; *n* = 1, 1.0%), which can be found in Table 2.

The majority of participants were female (*n* = 59, 58.4%). The mean age of participants was 37.9 years, ranging from 23 to 60 years old. Roles were grouped as registered nurses, physicians, nurse practitioners, and other (including nurse aids, care coordinators, allied health professionals, metal health emergency care team, and other) for analysis. Most participants were physicians (*n* = 53, 53%) or registered nurses (*n* = 26, 26%).

Hours worked per week ranged from 0 to 144, but 40 h was reported most often (*n* = 15, 15%). Of the 98 participants reporting their vaccine status, most (*n* = 85, 86.7%) reported they had received both doses of a COVID-19 vaccine, three (3.1%) participants received only a first dose, and ten (10.2%) reported not receiving any dose of the vaccine at the time of survey completion.

### 3.2. Stressful Life Events

The average life change index score among the participants was 281.0, and the average life change index count was 10.7. Nurse practitioners had both the highest average life change index score (335.9) and life change index count (13.1), compared to physicians and other roles. Healthcare workers in South America reported the highest average life change index score (583) as well as the highest average life change index count (19). The lowest average life change index score and average life change index count was for healthcare workers in Africa, which reported 0 for both.

Moreover, certain events were noted most frequently among participants as stressful. The change in number of family get-togethers occurred or was expected to occur among 84 participants (77%), followed by change in work hours or conditions (*n* = 75, 69%), and then change in social activities (*n* = 75, 69%). Those that answered yes to stressful life events and the weighted impact score of each event can be found in detail in Table 3.

### 3.3. Work Family Conflict and Work Life Balance

The average score among participants was 15.8. Registered nurses reported the highest average score of 16.4, compared to physicians with 15.4 and others with 16.1. Africa reported the highest average score of 17, followed by Australia and Europe with 16.5, then North America with 15.9, and Asia with 14.8.

### 3.4. Physical Symptoms

Responses among participants ranged from 13-52, and the average score was 27.1. Fatigue was the symptom reported most frequently, with 28.4% (*n* = 31) feeling fatigue every day and 33.9% (*n* = 37) reporting fatigue most days. Muscle pain was the second-highest symptom reported, with 14.6% (*n* = 16) reporting it every day and 13.7% (*n* = 15) reporting it most days. Backache was the third highest symptom, with 11.9% (*n* = 13) reporting it every day and 21.1% (*n* = 23) reporting it most days.

Registered nurses reported the highest frequency of the physical symptoms of fatigue (*n* = 10), muscle pain (*n* = 6), and backache (*n* = 5) every day. However, nurse practitioners reported the highest sum of physical symptoms on average (34). Moreover, 21.7% (*n* = 22) of females reported fatigue every day, compared to 6.9% (*n* = 7) of males. Those identifying as females also reported a higher amount of muscle pain (10.1%, *n* = 11) every day, and only 3.7% (*n* = 4) of males reported it every day. For backache, 9.2% (*n* = 10) of females reported every day, whereas 2.8% (*n* = 3) of males reported every day.

Participants from North America, Asia, and Europe experienced backache, fatigue, and muscle pain most frequently. For example, fatigue was experienced every day in North America (15.8%, *n* = 19), followed by 4.5% (*n* = 6) in Asia and 3.8% (*n* = 4) in Europe. It was found that 6.8% (*n* = 8) of participants from North America and 3.8% (*n* = 5) from Asia experienced backache every day. Muscle pain was experienced every day in North America with a reported 7.5% (*n* = 9), followed by Asia with 2.3% (*n* = 3), and then Europe with 2.3% (*n* = 3). However, on average, Asia reported the highest number of physical symptoms (30) compared to North America (26.9) and Europe (26.9). See Table 4 for more information on the experience of physical symptoms.

### 3.5. Linear Regression

A linear regression model indicated that the weighted stressful life changes score impacts physical symptom outcomes. For every one-unit increase in life change index score, it is estimated to increase physical symptoms by 9.9 (95% CI (6.94, 12.84), R^2^ = 0.29) (See Table 5).

Similarly, the frequency of stressful life events counted without weight indicated an impact on physical symptoms. For every one-unit increase in life change index count, it is estimated to increase physical symptoms by 0.3 (95% CI (0.24, 0.45), R^2^ = 0.28) (See Table 5). Another linear regression model indicated that the weighted stressful life changes score impacts work–family conflict. For every one-unit increase in life change index score, it is estimated to increase work–family conflict by 13.5 (CI 95% (6.01, 21.04), R^2^ = 0.29) (See Table 5). For every one-unit increase in life change index count, it is estimated to increase work–family conflict by 0.5 (CI 95% (0.25, 0.78), R^2^ = 0.12) (See Table 5).

## 4. Discussion

As a part of our study purpose, we gained more understanding for the additional stressors that emergency department healthcare workers experienced during the pandemic. For instance, the most frequently reported stressors included changes in family get-togethers, social activities, work hours or conditions, and sleeping habits. The highest counts and scores of stressful life events were found among nurse practitioners, a group not usually differentiated in research exploring the stress of emergency department workers [23]. Moreover, when examining the costs of this worker stress in hospitals, previous research has tended to focus on outcomes of burnout and job satisfaction [24]. However, it is also supported that stressors place an economic burden on organizations in general, costing American employers an estimated USD 300 billion every year in sick days, lost productivity, and associated costs for worker healthcare [25]. Unfortunately, the high life change index scores over 300 from participants in this study suggest that they are likely to become ill soon, and it is therefore an organizational concern. Certain negative health outcomes are known to be associated with stress. For instance, feelings of stress and burnout have generally been found to impact physical symptoms, increasing the risk for hyperlipidemia and type 2 diabetes, along with insomnia and depressive symptoms [26]. Our study also revealed a significant impact of stress on physical symptoms, highlighting the daily occurrence of fatigue, muscle pain, and backache among varying emergency department healthcare workers.

Further, protection policies may have increased the stress of the healthcare workers as they deal with the uncomfortable nature of wearing protection long-term and continuously [8] as well as the intense work demands that resulted from high volumes of patients who have a highly contagious and deadly virus that result high mortality [15]. The current literature supports that moderate-to-high levels of chronic fatigue and high levels of acute fatigue are generally experienced by hospital nurses [27], which is in line with our results indicating the highest frequency of fatigue experienced by registered nurses working in the emergency department. However, our results also add that nurse practitioners experienced the highest number of physical symptoms overall. Therefore, continued exploration is required in order to mitigate the physical risks placed on all emergency department healthcare workers.

Results from this study have suggested a lack of work–life balance among participants across roles and regions but particularly for registered nurses. When compared to a general sample of 3442 US adults in a previous study [11], physicians had reported significantly more dissatisfaction with work–life balance (40.2% vs. 23.2%). However, the present study found that registered nurses globally had the highest scores of work–family conflicts. These results have further revealed the potential impact on workers’ social and family lives, as an aspect of overall wellbeing.

Our study findings have uncovered certain considerations for hospitals to evaluate what they are doing in their emergency departments to promote work–life balance and improve health, by helping workers cope with workplace stress more effectively. This can be achieved through initiating support of peers or team members, support from oneself, and support from managers or organizations [28]. For example, peers in support groups may offer one another a sense of shared experience and offer strategies for effective coping [29]. Promoting team activities that focus on health may enhance worker wellbeing (e.g., a wellness challenge that includes fitness tracking). Further, encouraging self-care practices [30] and providing guidance on mindfulness strategies would help allow providers to settle their emotions through self-compassion and understanding, while also increasing the worker’s sense of control over their job [23]. Managers can work to cultivate a culture of health through ensuring that protective policies are in place that establish adequate staffing, reasonable workloads, and improved safety protocols. They can also continue to encourage workers and remind them that their efforts are seen and appreciated [23], as well as sharing mental health supports periodically [31].

The negative impacts on health, which are highlighted in our study, have already urged changes in the US legislation. The Dr. Lorna Breen Health Care Provider Protection Act (H.R.1667, S.610) has been introduced and passed by the House and the Senate to improve the health of healthcare workers and to prevent the consequences of stress and burnout [32]. Moreover, the International Labor Organization has previously released the first international standard for psychological safety and health: ISO 45001, Occupational health and safety management systems—Requirements with guidance for use [33]. This standard establishes the need to address and control cognitive and mental risks associated with work to reduce long-term effects on health.

### Limitations

There were important limitations of this study to address. First, the sample was collected through a convenience and snowball approach, which may have led to a less representative sample, potentially limiting generalizability [34]. This also required some countries to be grouped into regions. Second, there was a low number of participants in some regions, including only one participant from Africa, one from South America, and four from Australia. The participants were limited to those who self-selected themselves to participate after being sent the invitation. Third, the timing of study completion was a limitation, as COVID-19 cases were not necessarily in peak or in the beginning of the pandemic when there was more uncertainty. Instead, the participants were sampled more than a year after the pandemic began. We limited the survey time to 5 months (April to August 2021), which may have limited the number of responses. However, we wanted to have a more homogeneous COVID-19 exposure, as stresses change with infection rates. Moreover, the analysis was limited to participants without missing data, with a focus on the specific variables chosen; thus, there was a lack of ability to determine causality between stressors related to COVID-19 at this time. There were outliers in responses, including the participant in Africa noting the highest work–life conflict, yet the lowest stressful events. However, the results add valuable insight toward the need to make improvements for those working in emergency departments to enhance their work–life balance and health.

## 5. Conclusions

The overall results reveal high scores and counts of stressful life events among nurse practitioners, poor work–life balance among registered nurses, and a high number of physical symptoms among nurse practitioners working in the emergency department during the ongoing COVID-19 pandemic. The increase in stressful life events is significantly associated with both physical symptoms and work family conflict. There is a pressing need to consider improvements in the emergency department that better support their workforce and mitigate the risks associated with stress. Considerations for organizations include implementing protective policies and increasing support from managers, peers, and self, offering resources to enhance mental and physical health. Initiatives to care for those working on the frontline can continue to be advocated for during and beyond global pandemics to support a more sustainable workforce.

## Figures and Tables

**Figure 1 ijerph-19-03730-f001:**
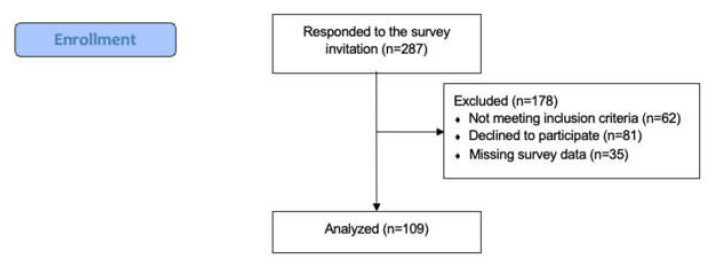
Study enrollment procedures.

**Table 1 ijerph-19-03730-t001:** Detailed demographics of emergency department workers that participated in the online survey investigating the impact of stressors during the COVID-19 pandemic.

Demographic Information		Count	Percentage
Gender	Male	41	40.6
	Female	59	58.4
Age (years old)	18–25	6	5.9
	26–35	38	37.6
	36–45	35	34.7
	46–55	18	17.8
	56 or older	4	4
Role	Physicians	53	53
	Registered Nurses	26	26
	Nurse Practitioners	7	7
	Nurse Aides	5	5
	Care Coordinators	1	1
	Allied Health Professionals	4	4
	Mental Health ECT	1	1
	Other	3	3
COVID-19 Vaccine Status	Fully vaccinated	85	86.7
	One dose	3	3.1
	Not vaccinated	10	10.2
Country	Australia	4	4.1
	Canada	5	5.1
	China	13	13.3
	Colombia	1	1
	India	1	1
	Mexico	1	1
	Pakistan	1	1
	Philippines	1	1
	Qatar	1	1
	Saudi Arabia	2	2
	Slovenia	1	1
	Tanzania	1	1
	Turkey	1	1
	United Kingdom	10	10.2
	United States	55	56.1
Hours Worked per Week	≤20	12	12
	21 to 40	55	55
	41 to 60	24	24
	61 to 80	5	5
	≥81	4	4

**Table 2 ijerph-19-03730-t002:** Condensed demographics used for data analysis to determine if there was a correlation between stressors and wellbeing during the COVID-19 pandemic.

Condensed Demographic Information		Count	Percentage
Role	Physicians	53	53
	Registered Nurses	26	26
	Nurse Practitioners	7	7
	Other	14	14
Region	North America	61	62.2
	Asia	20	20.4
	Europe	11	11.2
	South America	1	1
	Australia	4	4.1
	Africa	1	1

**Table 3 ijerph-19-03730-t003:** Stressful life events reported by emergency department workers during the COVID-19 pandemic.

Stressful Life Event	Event Impact Score	Number of Participants Who Answered Yes	Percentage of Participants Who Answered Yes (*n* = 109)
Death of a Spouse	100	1	1%
Divorce	73	3	3%
Marital Separation	65	2	2%
Jail Term	63	1	1%
Death of Close Family Member	63	20	18%
Personal Injury or Illness	63	22	20%
Marriage	50	7	6%
Fired at Work	47	4	4%
Marital Reconciliation	45	5	5%
Retirement	45	1	1%
Change in Health of Family Member	44	40	37%
Pregnancy	40	7	6%
Sex Difficulties	39	37	34%
Gain of a New Family Member	39	18	17%
Business Readjustment	39	18	17%
Change in Financial Sate	38	37	34%
Death of a Close Friend	37	15	14%
Change to a Different Line of Work	36	18	17%
Change in Number of Arguments with Spouse	35	41	38%
Mortgage over $20,000	31	44	40%
Foreclosure of Mortgage or Loan	30	4	4%
Change in Responsibilities at Work	29	67	61%
Son or Daughter Leaving Home	29	8	7%
Trouble with In-Laws	29	13	12%
Outstanding Personal Achievement	28	26	24%
Spouse Begins or Stops Work	26	25	23%
Begin or End School	26	18	17%
Change in Living Conditions	25	37	34%
Revisions of Personal Habits	24	54	50%
Trouble with Boss	23	23	21%
Change in Work Hours or Conditions	20	75	69%
Change in Residence	20	31	28%
Change in Schools	20	2	2%
Change in Recreational Activities	19	41	38%
Change in Church Activities	19	23	21%
Change in Social Activities	19	75	69%
Mortgage or Loan less than $20,000	17	12	11%
Change in Sleeping Habits	16	73	67%
Change in Number of Family Get-togethers	15	84	77%
Change in Eating Habits	15	67	61%
Vacation	13	46	42%
Christmas Approaching	12	22	20%
Minor Violation of the Law	11	2	2%

**Table 4 ijerph-19-03730-t004:** Physical symptoms reported by emergency department workers during the COVID-19 pandemic.

Symptom	Everyday	Most Days	Once or Twice per Week	Once or Twice	Not At All
Upset stomach or nausea (*n* = 109)	0	7.34%, *n* = 8	33.94%, *n* = 377	26.61%, *n* = 29	32.11%, *n* = 35
Backache (*n* = 109)	11.93%, *n* = 13	21.10%, *n* = 23	19.27%, *n* = 21	24.77%, *n* = 27	22.94%, *n* = 25
Headache (*n* = 109)	6.42%, *n* = 7	19.27%, *n* = 21	25.69%, *n* = 28	32.11%, *n* = 35	16.5%, *n* = 18
Acid indigestion or heartburn (*n* = 109)	3.67%, *n* = 4	13.76%, *n* = 15	21.10%, *n* = 23	27.52%, *n* = 30	33.94%, *n* = 37
Diarrhea (*n* = 109)	0.93%, *n* = 1	3.70%, *n* = 4	11.11%, *n* = 12	28.70%, *n* = 31	55.56%, *n* = 60
Stomach cramps (non-menstrual) (*n* = 109)	0.00%, *n* = 0	3.67%, *n* = 4	16.51%, *n* = 18	25.69%, *n* = 28	54.13%, *n* = 59
Loss of appetite (*n* = 109)	0.00%, *n* = 0	9.17%, *n* = 10	16.51%, *n* = 18	26.61%, *n* = 29	47.71%, *n* = 52
Shortness of breath/difficulty breathing (*n* = 109)	0	2.75%, *n* = 3	8.26%, *n* = 9	14.68%, *n* = 16	74.31%, *n* = 81
Dizziness (*n* = 109)	0.00%, *n* = 0	2.75%, *n* = 3	13.76%, *n* = 15	19.27%, *n* = 21	64.22%, *n* = 70
Chest pain (*n* = 109)	0.92%, *n* = 1	4.59%, *n* = 5	8.26%, *n* = 9	15.60%, *n* = 17	70.64%, *n* = 77
Flu or cold symptoms (fever, sore throat, chills) (*n* = 109)	0	2.75%, *n* = 3	6.42%, *n* = 7	22.02%, *n* = 24	68.81%, *n* = 75
Muscle pain (*n* = 109)	13.76%, *n* = 15	14.68%, *n* = 16	13.76%, *n* = 15	24.77%, *n* = 27	33.03%, *n* = 36
Tiredness or fatigue (*n* = 127)	28.44%, *n* = 31	33.94%, *n* = 37	20.18%, *n* = 22	9.17%, *n* = 10	8.26%, *n* = 9

**Table 5 ijerph-19-03730-t005:** Linear regression results for life change index count and sum of physical symptoms with sum of physical symptoms and work life balance.

Sum of Physical Symptoms ^^^				
**Life Change Index Score**	**Estimate**	**Std Error**	**t Ratio**	**Prob > |t|**
Intercept	13.10551	42.286	0.31	0.7572
Sum of Physical Symptoms	9.892446	1.487464	6.65	<0.0001 *
**Life Change Count**	**Estimate**	**Std Error**	**t Ratio**	**Prob > |t|**
Intercept	1.357206	1.521613	0.89	0.3744
Sum of Physical Symptoms	0.34555	0.053525	6.46	<0.0001 *
**Work Life Balance**				
**Life Change Index Score**	**Estimate**	**Std Error**	**t Ratio**	**Prob > |t|**
Intercept	67.86831	61.46494	1.1	0.272
Sum for work life balance	13.52349	3.790421	3.57	0.0005 *
**Life Change Count**	**Estimate**	**Std Error**	**t Ratio**	**Prob > |t|**
Intercept	2.588992	2.174426	1.19	0.2364
Sum for work life balance	0.5156	0.134093	3.85	0.0002 *

^ t = standard *t*-test value, * significance level < 0.05.

## Data Availability

Data sharing is not applicable to this article.

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
