# Peer review of "Examining the Impact of Stressors during COVID-19 on Emergency Department Healthcare Workers: An International Perspective"

_ijerph, 2022, doi:10.3390/ijerph19063730_

Round 1

Reviewer 1 Report

This interesting study aimed to examine the impact of stressors among emergency department healthcare workers in a global perspective. The authors made a description of stressful events that occurred during the COVID-19 pandemic among emergency department healthcare workers, investigating their impact on physical health and emotions. The study intended to analyse epidemiological data on this subject "across the globe" based on the total worker health initiative.

The study has an interesting purpose and valuable insights in general. On the other hand, it also raise some concerns about its justification and methods.

As the authors mention in the introduction, several studies have already addressed similar research questions, evaluating the role of occupational factors and other stressors among healthcare workers during the COVID-19 pandemic. Therefore, how and why the authors believe their study can contribute with the literature? 

Studying this topic in an international perspective, with representative samples of different groups of healthcare workers across the globe could be really interesting. However, the authors used a restrict convenience sample, not representative. In several countries only 1 or very few people individuals participated in the study. With all limitations concerning this type of sample selection, I would expect that the use of the "snowball" method in 15 countries would provide a wider participation. Why did the authors not expanded the period of data collection? Is this sample enough to answer the research question? Also, a deeper explanation on the selection of participants would be valuable. Why only 287 were invited? Why so many people who were invited did not meet inclusion criteria or refused to participate?

The measurements and instruments used by the authors seem to be valid, but some of them did not have their operationalization described. Why did the authors used the sum of symptoms as an outcome? Was this method validated previously? Without this information, it is not clear whether the use of linear regression is appropriate or not in this case.

Results are very descriptive and simple. A table with the main results (from the linear regression) might be interesting.

The discussion is short and repeats results in a few sentences. More literature can be added to the discussion, enriching authors´arguments. Sometimes is not clear the references used were related to emergency department workers or healthcare workers and nurses in general. Some of the references are only related to nurses. A more specific focus on the target population and its particularities is necessary.

More limitations should be discussed. Also, considering the limitations of the study, I believe some recommendations and conclusions based on the results should be softned.

Author Response

Please see the attachmen.

Reviewer 2 Report

  • line 41, Measures to prevent healthcare professionals from contracting severe acute respiratory syndrome during high-risk surgical procedures have been taken and disseminated among healthcare professionals. However, these measures are often astringent and difficult for the worker who is often subjected to enormous stress overload. please cite doi:10.1007/s10096-003-1068-2
  • line 50, moreover, the mandatory use of protective devices has led to an increase in the discomfort of the health worker, who often reported rhinitis symptoms, poor concentration due to the device. However, the application of simple comfort rules could alleviate this symptomatology. please cite doi:10.7416/ai.2021.2439
  • line 77, The SARS-CoV2 pandemic has put a strain on healthcare systems around the world. The high volume of patients, combined with an increased need for intensive care and potential transmission, has forced the reorganization of hospitals and care delivery models. In this article, approaches to minimize the risk to ENTs during their COVID-19 infected patients are presented. Standard operating procedures have been adapted for both facilities and healthcare professionals, including the development of well-defined and separate patient care areas for the treatment of those affected by COVID-19. The availability of personal protective equipment (PPE) and adequate training of health professionals on their use must be ensured. Preventive measures are especially important in Otolaryngology-Head and Neck Surgery, as exposure to saliva, droplet, and aerosol suspensions are increased in routine upper aero-digestive tract examination. Additionally, frequent invasive procedures, such as laryngoscopy, intubation, or tracheostomy placement and care, pose a high risk of contracting COVID-19. please cite doi:10.23750/abm.v92i1.11281
  • line 104, could you add a code number?
  • line 113 according which certification?
  • line 127 cite the referecens. why this tool?
  • line 158, Kolmogorov smirnov was perfromed?
  • line 271, The SARS-CoV2 pandemic has put a strain on healthcare systems around the world. The high volume of patients, combined with an increased need for intensive care and potential transmission, has forced the reorganization of hospitals and care delivery models. In this article, approaches to minimize the risk to ENTs during their COVID-19 infected patients are presented. Standard operating procedures have been adapted for both facilities and healthcare professionals, including the development of well-defined and separate patient care areas for the treatment of those affected by COVID-19. The availability of personal protective equipment (PPE) and adequate training of health professionals on their use must be ensured. Preventive measures are especially important in Otolaryngology-Head and Neck Surgery, as exposure to saliva, droplet and aerosol suspensions is increased in routine upper aero-digestive tract examination. Additionally, frequent invasive procedures, such as laryngoscopy, intubation, or tracheostomy placement and care, pose a high risk of contracting COVID-19. please cite doi:10.23750/abm.v92i1.11281
  • A flow diagram with the study protocol and criteria should be included
  • a figure or table with teh survey was added?
  •  

Reviewer 3 Report

Few respondents were included in the study in some regions of the world to thoroughly examine the impact of stressors during COVID-19 on health professionals globally. Nevertheless, I rate the paper as beneficial also with regard to the contribution of the study to the change in US legislation.

  1. How did the results of the study contribute to improving the working conditions of the healthcare workers in the other studied regions of the world?
  2. Why did the authors not include more healthcare professionals from Europe (eg Italy) in the study? Europe has been hit hard by the COVID-19 pandemic.
